# Monitoring and Prediction of Glacier Deformation in the Meili Snow Mountain Based on InSAR Technology and GA-BP Neural Network Algorithm

**DOI:** 10.3390/s22218350

**Published:** 2022-10-31

**Authors:** Zhengrong Yang, Wenfei Xi, Zhiquan Yang, Zhengtao Shi, Tanghui Qian

**Affiliations:** 1Faculty of Geography, Yunnan Normal University, Kunming 650500, China; 2Key Laboratory of Early Rapid Identification, Prevention and Control of Geological Diseases in Traffic Corridor of High Intensity Earthquake Mountainous Area of Yunnan Province, Kunming 650093, China; 3Faculty of Public Safety and Emergency Management, Kunming University of Science and Technology, Kunming 650093, China; 4Key Laboratory of Geological Disaster Risk Prevention and Control and Emergency Disaster Reduction of Ministry of Emergency Management of the People’s Republic of China, Kunming University of Science and Technology, Kunming 650093, China

**Keywords:** InSAR technology, mountain glaciers, glacier deformation, GA-BP, monitoring and prediction

## Abstract

The morphological changes in mountain glaciers are effective in indicating the environmental climate change in the alpine ice sheet. Aiming at the problems of single monitoring index and low prediction accuracy of mountain glacier deformation at present, this study takes Meili Mountain glacier in western China as the research object and uses InSAR technology to construct the mountain glacier deformation time series and 3D deformation field from January 2020 to December 2021. The relationship between glacier deformation and elevation, slope, aspect, glacier albedo, surface organic carbon content, and rainfall was revealed by grey correlation analysis. The GA-BP neural network prediction model is established from the perspective of multiple factors to predict the deformation of Meili Mountain glacier. The results showed that: The deformation of Meili Mountain glacier has obvious characteristics of spatio-temporal differentiation; the cumulative maximum deformation quantity of glaciers in the study period is −212.16 mm. After three-dimensional decomposition, the maximum deformation quantity of glaciers in vertical direction, north–south direction and east–west direction is −125.63 mm, −77.03 mm, and 107.98 mm, respectively. The average annual deformation rate is between −94.62 and 75.96 mm/year. The deformation of Meili Mountain glacier has a gradient effect, the absolute value of deformation quantity is larger when the elevation is below 4500 m, and the absolute value of deformation quantity is smaller when it is above 4500 m. The R^2^, MAPE, and RMSE of the GA-BP neural network to predict the deformation of Meili glacier are 0.86, 1.12%, and 10.38 mm, respectively. Compared with the standard BP algorithm, the prediction accuracy of the GA-BP neural network is significantly improved, and it can be used to predict the deformation of mountain glaciers.

## 1. Introduction

Mountain glaciers are glaciers that develop in snowy areas above the snowline in low to mid-latitude alpine regions and flow slowly downhill along mountain slopes or troughs under the effect of gravity [1,2]. Mountain glaciers are usually characterized by higher snowline positions, thinner ice, smaller size, higher velocity, and energy of movement, as well as stronger ice erosion due to the rigorous control of surface morphology in alpine areas [3]. Mountain glaciers make up only about 1% of the world’s ice, compared with the Antarctic and Greenland ice sheets (>99%), but their changing patterns have dominated the rise and fall in sea level in the 20th century [4]. China has the largest number of mountain glaciers in the world, and most of its glaciers are mountain glaciers [5]. In addition, mountain glaciers have irreplaceable ecological service functions for the population, resource environment, and economic development of agriculture and animal husbandry in arid and semi-arid regions for their special spatial and temporal distribution and evolutionary processes [6,7]. Mountain glaciers are in a state of rapid melting in the context of global warming [8]. The results of China’s Second Glacier Inventory suggest that the glacier area in western China has shrunk by 18% since 1950, of which mountain glaciers account for over half [9]. Melting glaciers change the regional ecology and raise the global sea level while inducing a series of glacial disasters that affect human public health and safety [6,10,11]. Thus, monitoring mountain glacier deformation trends, analyzing the factors for glacier displacement changes, and predicting glacier deformation take on a critical significance in regional ecological protection and human public health safety.

Conventional means of mountain glacier monitoring are usually performed through field survey measurements, including field stereo photogrammetry [12], Global Navigation Satellite System (GNSS) technology [13], LiDAR, and ground-penetrating radar measurement techniques [14]. On that basis, the glacier surface elevation of monitoring points can be directly measured, and monitoring results with high accuracy can be obtained. However, the effective range of temperate glacier monitoring by conventional means is less than 1% due to the extreme climatic conditions and complex topography of the alpine ice sheet environment [15]. Currently, optical remote sensing is one of the critical tools adopted to generate regional inventories of mountain glaciers and extract glacier boundaries and area changes [16,17], whereas it is susceptible to natural cloud conditions, especially under the extreme climatic conditions of the ice sheet environment in alpine regions. Moreover, the effectiveness of optical remote sensing for mountain glacier monitoring has been poor [15].

Interferometric Synthetic Aperture Radar (InSAR) technology has been used for conventional field surveys and optical remote sensing techniques over the past few years. InSAR technology is based on active synthetic aperture radar (SAR) technology, which shows the advantages of a large range, all-day, all-weather, good stability, strong dynamics, high accuracy, as well as high resolution [18]. It has been extensively employed in urban surface subsidence [19,20], seismic activity [21,22], volcanic hazards [23,24], landslide hazards [25,26,27], as well as glacier displacement monitoring [15,16,17,18]. The earliest application in surface deformation monitoring is the differential InSAR (D-InSAR) technique, which has been well-adopted to invert glacier surface displacements under short-time baselines [28]. However, the change in scattering characteristics during glacier physical transformation may cause temporal and spatial decoherence phenomena due to its ability to detect relative deformation between two-time phases only, and the extreme complexity of the alpine ice sheet environment, thus significantly affecting the accuracy of monitoring results [29]. Multi-temporal InSAR technology has been proposed (e.g., persistent scatterer InSAR (PS-InSAR) [30], small baseline subset InSAR (SBAS-InSAR) [7], and other time series InSAR technology) to reduce the D-InSAR technology’s susceptibility to spatio-temporal solution correlation, geometric solution correlation, and atmospheric delay. Temporal InSAR technology employs multi-temporal radar data observed by repeated orbits to provide stable and reliable phase observation targets through detection in the interferogram and the analysis of the temporal phase of interference points under the winding condition to retrieve the temporal displacement and deformation information of mountain glaciers [31]. In addition, the radar line of sight (LOS) deformation obtained by the InSAR technique indicates the geometrically transformed projection of the surface deformation vector in the LOS direction, instead of the true surface deformation, which is due to the side-looking radar imaging method [32]. Glacier morphology change is a complex process, which comprises glacier area, glacier surface elevation, slope, thickness, debris thickness, ice cliff size, and number, etc. To be specific, the change in glacier end takes on a critical role in the glacier morphology change [7]. Azimuth Pixel Offset (AZO) [33] and Multiple Aperture Interference (MAI) [34] have been proposed to build a 3D deformation field of surface deformation. The technique has been proposed to fuse the ascending and descending orbit D-InSAR and azimuthal AZO or MAI deformation results to build a 3D deformation field of mountain glaciers and obtain the morphological change components of glaciers in vertical, east–west, and north–south directions, such that a more comprehensive insight is provided into glacier morphological changes compared with the line-of-sight deformation results. The current applications of InSAR technology in glaciers primarily have the following aspects: (1) monitoring glacier deformation using D-InSAR technology [16,35,36]; (2) obtaining time series of glacier deformation using MT-InSAR [6,7,15,17,28]; (3) and fusing LOS and AZI directional deformation information to build glacier 3D deformation fields [34,37,38]. The above methods confirm that the InSAR technique can effectively acquire glacier deformation information in extremely complex ice sheet environments, and it has greater potential for application in monitoring glacier flow velocity and morphology.

Mountain glaciers exhibit significant response characteristics to climate and environmental changes [1], and combining climate and environmental factors to explore the drivers of glacier deformation and predict its deformation trend takes on a great significance in studying global climate change and glacier disaster warning. Conventional deformation prediction models are divided into three main categories: (1) physical process models [39], modelling and simulating the deformation process based on the physical mechanism of deformation; (2) mathematical and statistical models [39], using discrete time series data to predict the future deformation state; (3) neural network models 40, using the obtained historical deformation data to build neural networks to predict future deformation values. However, the physical parameters of mountain glacier deformation are extremely complex, of which data are difficult to obtain [7], and the application of physical process models has been severely limited. Mathematical and statistical models often lack a physical and geoscientific basis [39], and simple statistical laws cannot explain and predict complex deformation of mountain glaciers. Benefiting from the rapid development of computer performance and artificial intelligence, considerable successful cases have been achieved in the application of artificial neural networks in the field of prediction [40]. To be specific, the BP neural network based on the error back-propagation training algorithm can approximate any continuous functions and have a strong nonlinear mapping capability [41], which provides new technical support for fast and nonlinear glacier deformation [15] prediction. However, the conventional BP neural network has many parameters and should update more thresholds and weights, resulting in slower convergence, longer training time, and lower prediction accuracy, and the BP model utilizes a gradient descent optimization algorithm, which is prone to fall into local minima [41]. Accordingly, it is highly challenging to optimize the conventional BP neural network model and make it accurate and effective in predicting mountain glacier deformation in complex ice sheet environments.

Thus, to solve the limitations in the monitoring and prediction of mountain glacier deformation, this study proposed a method that combines InSAR technology and Genetic Algorithm optimized Back-Propagation (GA-BP) neural network to monitor and predict mountain glacier deformation and built a prediction model from a multi-factor perspective. The spatial distribution and temporal evolution of mountain glacier morphological changes were analyzed, and the relationships between mountain glacier morphological variables and climate and environmental factors were also developed.

## 2. Materials and Methods

### 2.1. Study Area

The study area is located at the border between the eastern part of Zayü County in the Tibet Autonomous Region and the western part of Deqin County in Yunnan Province (98°33′23.52″–98°49′16.68″ E, 28°2′54.79″–28°33′1.37″ N) (Figure 1). The study area is situated in the Nu Mountain Range on the Yunnan–Tibet border between the Lancang and Nu Rivers. The southeastern edge of the Qinghai-Tibet Plateau refers to a transition zone between the first and second terrain steps in China [42]. It takes up a total area of nearly 1000 km², and the stratigraphic outcrops are primarily Palaeozoic–Mesozoic strata, including Devonian, Carboniferous, Permian, Triassic, Jurassic, White Malaise, and a few Paleozoic and Quaternary strata, which are narrowed by the sharp compression of the arc of the Sanjiang-Indian fold system in the ancient trough. The topography is high in the north and low in the south, with strong fracture activity [43]. The area has the highest altitude in the area as 6740 m at Kawagbo Peak and the lowest altitude of 2038 m at the junction of the Lancang River at the foot of the mountain, with an altitude difference of 4702 m. It is characterized by a cold temperate mountainous climate.

The Meili Mountain glacier area is characterized by high slopes, long gorges, and deep valleys and it is controlled by the alternating southwest Indian monsoon and temperate prevailing westerly winds, with warm and humid air currents penetrating from south to north from the three parallel river valleys of the Jinsha, Nu, and Lancang rivers, thus resulting in the formation of mountain glaciers of enormous size and abundant precipitation in the area [42]. The Second Glacier Inventory Dataset of China (V1.0) [9] suggests that there have been 55 glaciers in the study area, with a total glacier area of nearly 128.54 (Figure 2a). The average elevation of mountain glaciers in the study area ranges from 4600 m to 5600 m (Figure 2b–d), and mountain glaciers have been widely developed on slopes between 16° and 38° (Figure 2e), with average aspects ranging from 16° to 355° (Figure 2f).

### 2.2. Data

#### 2.2.1. SAR Data

In this study, Sentinel-1 data acquired by the European Space Agency (ESA) were employed as the primary dataset to monitor trends in glacier melting, accessed online at https://scihub.copernicus.eu/dhus/#/home (accessed on 30 April 2022). The Sentinel-1 satellite was launched on 3 April 2014 and comprised a constellation of two polar-orbiting satellites, Sentinel-1A and Sentinel-1B. Sentinel-1 satellite carried a C-band synthetic aperture radar (SAR) with a right-viewing active phased array antenna at a wavelength of 5.6 cm, operating at a central frequency of 5.405 GHz, thus providing Stripmap (SM), Interferometric Wide swath (IW), as well as a satellite with a frequency of 5.405 GHz. Interferometric Wide swath (IW), Extra Wide swath (EW), and Wave (WV) represent four imaging modes to provide all-day, all-weather radar imagery of the land and ocean [44].

A total of 50 S1 TOPS modes single-look complex (SLC) images in Interferometric Wide swath (IW) mode, polarized in VV, with an amplitude of 250 Km, were selected for this study since horizontal polarization (VV) exhibits a stronger backscatter intensity than cross-polarization (VH) in glacial regions [7] and the size of the study area. The distance and azimuthal resolutions were 5 m and 20 m, respectively, and the angles of incidence of the elevation data were 39.27° and 39.51°, respectively. The selected Sentinel-1 ascending orbit data were acquired at 11 a.m. and descending orbit data were acquired at 23 p.m. In order to establish the relationship between the environmental climate and environmental factors of alpine ice sheets and the deformation quantity of mountain glaciers, 48 descending orbit image datasets spanning from 9 January 2020 to 29 December 2021 were used as monitoring data to construct the deformation time series of mountain glaciers. The other two ascending orbit images on 7 January 2020 and 27 December 2020 were used as auxiliary data to obtain the three-dimensional deformation of the glacier.

#### 2.2.2. Glacier Deformation Impact Factors and Other Data

The main climatic and environmental factors for mountain glacier deformation involved elevation, slope, slope orientation, glacier albedo, surface organic carbon content and rainfall in the study area. Elevation, slope, and aspect were obtained using ArcGIS 10.8 software with a DEM of the study area as the input data. The DEM data were acquired using a Digital Elevation Model (DEM) with 30 m spatial resolution from JAXA ALOS WORLD 3D, 30 m spatial resolution, to calculate elevation, slope, and aspect and remove the effects of topographic phase, available online at: https://www.eorc.jaxa.jp/ALOS/en/aw3d30/data/index.htm (accessed on 1 May 2022). Glacier albedo data were obtained using Landsat 8 data based on the GEE platform to obtain blue band, red band, near-infrared band, shortwave infrared 1, as well as shortwave infrared 2. Soil organic carbon content data, available online from FAO: http://54.229.242.119/GSOCmap/ (accessed 10 May 2022). Rainfall data, available online from Rainfall Processing Systems: https://arthurhou.pps.eosdis.nasa.gov (accessed 15 May 2022).

Ancillary data consisted of the China Second Glacier Inventory dataset (V1.0), available online from the National Glacial Permafrost Desert Science Data Centre at http://www.ncdc.ac.cn/portal/ (accessed 20 April 2022). High-resolution Google images (Google Earth) are available online at http://www.google.cn/intl/zh-CN/earth/ (accessed 20 May 2022). The data acquired from Google Earth were employed for positional annotation of the Meili Mountain glacier dataset and for overlaying the acquired InSAR deformation results for analysis. Table 1 lists specific data and specifications.

## 3. Methodology and Data Processing

Figure 3 illustrates the overall technical process of this study. First, the SBAS-InSAR technique, D-InSAR technique, and MAI technique were adopted to build the time series deformation and 3D deformation fields of the Meili Mountain glacier from January 2020 to December 2021, to identify the melting and accumulation areas of the Meili Mountain glacier, as well as to analyze the spatial distribution characteristics and temporal evolution pattern of the morphological changes in the mountain glacier. Second, the effect of altitude, slope, aspect, glacier albedo, surface organic carbon density, and rainfall on glacier morphological variables was analyzed using grey correlation analysis. Lastly, a GA-BP neural network model was built to predict the morphological trends of Meili Snow Mountain glaciers from a multi-factor perspective.

### 3.1. Acquisition of 3D Surface Deformation Information by D-InSAR and MAI Techniques

Fusion of D-InSAR and MAI has been widely used to acquire surface displacement components and create 3D deformation fields in accordance with SAR image phase information [32]. Based on the four ascending and descending orbit images of January 2020 and December 2021 in the study area, with the SARScape5.2.1, the same orbits were employed as the primary and secondary images, respectively, in chronological order. Moreover, the baseline estimation was performed on the single-look complex images, and the image alignment was performed in accordance with the imaging geometry relationship between the two. Next, the multi-view number was set to 4:1 to more effectively reduce speckle noise. The interferometric workflow was performed using Minimum Cost Flow decoupling method and Goldstein filtering method, and the final interferometric phase map was generated, at which point the interferometric phase is expressed in Equation (1).
(1)$$\varphi =-\frac{4\pi }{\lambda }\left(R_{1}-R_{2}\right)=\varphi_{flat}+\varphi_{topo}+\varphi_{def}+\varphi_{noise}+\varphi_{atm}$$
where $R_{1},\;R_{2}$ denotes the distance from the sensor to the ground point; $\varphi_{flat}$ represents the flat earth phase; $\varphi_{topo}$ is the terrain phase; $\varphi_{def}$ is the deformation phase; $\varphi_{noise}$ expresses the noise phase, and $\varphi_{atm}$ is the phase due to atmospheric delay. ALOS WORLD 3D 30 m DEM was adopted to obtain the interferometric phase stripe map of the study area. The surface deformation phase of the study area was obtained after the secondary differential processing to remove the flat earth effect. Subsequently, the surface deformation variable in the radar line-of-sight direction of the study area was obtained (Figure 4a,b). The coherence of primary and secondary images are shown in Figure 4c,d.

The MAI technique is capable of generating forward and backward interferograms through sunbeam decomposition in the SAR azimuthal direction under the Doppler positive and negative conditions of the radar echoes. Subsequently, it obtains the azimuthal displacement component of the orbit in accordance with the phase difference of the interferograms [32]. Using the SARScape5.2.1, the interferograms were generated by selecting the primary and secondary images from the four images used in DInSAR described in the previous section. Band-pass filtering was performed on the primary and secondary images to obtain the forward and backward single-look complex images of the primary and secondary images, respectively. The MAI interferometric phase was obtained through conjugate multiplication of the front and rear-view phases using conventional interferometric methods, as expressed in Equation (2).
(2)$$\varphi_{MAI}=\varphi_{f}-\varphi_{b}=-\frac{4\pi }{l}nx$$
where x denotes the azimuthal deformation; l is the effective antenna length; and n represents the normalized oblique viewing angle coefficient. The azimuthal deformation of the study area was obtained after Goldstein filtering and area growth method of decoupling, correction for orbital errors and atmospheric delays, orbital refinement, re-deflating, as well as geocoding (Figure 5a,b), and the coherence of primary and secondary images are shown in Figure 5c,d.

In accordance with the imaging geometry of the SAR satellite, it is assumed that the 3D deformation components of the glacier in the vertical, north–south, and east–west directions are $d={\left[d_{u},d_{n},d_{e}\right]}^{T}$. Subsequently, the relationship between the D-InSAR or MAI results described above and the 3D deformation component results is expressed in Equation (3).
(3)R=uD
where u denotes the coefficient matrix of the observation equation, related to the radar incidence angle θ and the orbital azimuth angle α. The radar incidence angles $\theta_{A}$ and $\theta_{D}$ and the orbital azimuth angles $\alpha_{A}$ and $\alpha_{D}$ of the ascending and descending orbit data are substituted into the above equation to yield Equation (4).
(4)$$\left[\begin{array}{c} R_{los}^{A}\\ R_{los}^{D}\\ R_{azi}^{A}\\ R_{azi}^{D} \end{array}\right]=\left[\begin{array}{ccc} \cos\theta_{A} & \sin\alpha_{A}\sin\theta_{A} & -\cos\alpha_{A}\sin\theta_{A}\\ \cos\theta_{D} & \sin\alpha_{D}\sin\theta_{D} & -\cos\alpha_{D}\sin\theta_{D}\\ 0 & \cos\alpha_{A} & \sin\alpha_{A}\\ 0 & \cos\alpha_{D} & \sin\alpha_{D} \end{array}\right]\cdot \left[\begin{array}{c} d_{u}\\ d_{n}\\ d_{e} \end{array}\right]$$

In Equation (4), a total of four observations in the line-of-sight and azimuthal directions are obtained using D-InSAR and MAI techniques, and three unknown components of 3D deformation in the vertical, north–south, and east–west directions are obtained, which are easily derived from the weighted least squares principle [34], as expressed in Equation (5).
(5)$$D=-{\left(u^{T}Q^{-1}u\right)}^{-1}\cdot u^{T}Q^{-1}R$$
where R denotes the observation matrix; u expresses the coefficient matrix of the observation equation; and Q represents the observation equation weight matrix. Since the pre-test variance of the D-InSAR and MAI techniques is unknown, the accuracy of the acquired line-of-sight and azimuthal deformation variables is known to be of the same centimeter level, the observation equation weight matrix is processed with equal weights and solved using matlab2020a software to obtain the 3D deformation components of the Meili Mountain glacier, as illustrated in Figure 6.

### 3.2. SBAS-InSAR Ground Surface Time Series Deformation Information Acquisition

The acquired 3D morphological variables of mountain glaciers cannot be analyzed in time series for glacier melting trends since they only detect displacement changes between two-time phases. Accordingly, the SBAS-InSAR technique was introduced, in the SARScape5.2.1, and the 48 Sentinel-1A single-look complex images of the study area from 9 January 2020 to 29 December 2021 were selected, with the image dated 9 March 2020 as the super master image. A temporal baseline threshold of 60 d was set, and the spatial baseline was set to 45% of the critical baseline threshold. On that basis, a total of 292 interferometric image pairs were generated. To suppress speckle noise, the multi-view number was set to 1:4, and the interferometric processing was performed using the Minimum Cost Flow decoupling method and the Goldstein filtering method. The combined interferometric pairs were aligned and adjusted to remove undesirable data to produce an interferogram, at which point the interferometric phase is expressed in Equations (6) and (7).
(6)$$\Delta \varphi_{i}=\varphi_{t_{1}}-\varphi_{t_{2}}\approx \Delta \varphi_{i_{def}}+\Delta \varphi_{i_{topo}}+\Delta \varphi_{i_{atm}}+\Delta \varphi_{i_{noise}}+\Delta \varphi_{i_{orb}}$$
(7)$$\left\{ \begin{array}{l} \Delta \varphi_{i_{def}}\left(x,r\right)=\frac{4\pi }{\lambda }\left[d\left(t_{2}\right)-d\left(t_{2}\right)\right],i=1,2,\cdots ,m\\ \Delta \varphi_{i_{topo}}\left(x,r\right)=\frac{4\pi }{\lambda }\cdot \frac{B_{⊥}\Delta h}{r\sin\theta }\\ \Delta \varphi_{i_{atm}}\left(x,r\right)=\varphi_{atm}\left(t_{2}\right)-\varphi \left(t_{1}\right) \end{array}\right.$$
where $\Delta \varphi_{i_{def}}$ denotes the phase from the slope-to-distance deformation; $\Delta \varphi_{i_{topo}}$ represents the topographic phase; $\Delta \varphi_{i_{atm}}$ is the phase due to atmospheric delay; $\Delta \varphi_{i_{noise}}$ represents the phase due to coherent noise, and $\Delta \varphi_{i_{orb}}$ is the phase caused by orbital uncertainty. GCP points (Figure 7a) without redundant terrain streaks and phase jumps and far away from the deformation region were selected as the reference region for InSAR measurement. After orbit refining and re-deflating, the rank-deficient matrix B of m×n is obtained, and the matrix equation is shown in Equation (8):(8)Bv=δφ

The least square method and singular value matrix decomposition were used for deformation inversion, followed by estimation and removal of atmospheric phases to obtain time series deformation information for the study area, as presented in Figure 7b,c. The average coherence of SAR images selected in the study period was shown in Figure 7d.

### 3.3. GA-BP Neural Network Model Construction and Glacier Deformation Prediction

Genetic algorithms (GAs) are stochastic global search and optimization methods developed to simulate the mechanisms of biological evolution in nature. GA use Darwinian evolution and Mendelian genetics to automatically acquire and accumulate knowledge regarding the search space during the search process and to efficiently parallel and adaptively control the global search to yield the optimal solution [41].

The Back-Propagation (BP) neural network refers to a multi-layer feed-forward neural network trained by the error back-propagation algorithm, which learns and stores considerable “input–output” pattern mapping relationships [41]. The BP model comprises an input layer, an implicit layer, and an output layer. Its core is to develop the functional relationship between input samples and expected values using the implicit form of the neural network. The BP neural network exhibits the ability of arbitrary complex pattern classification and excellent multi-dimensional function mapping ability, as well as strong non-linear mapping ability and flexible network structure. It has been adopted to achieve effective results in surface subsidence monitoring and prediction.

Combining the GA algorithm and BP neural network can overcome the problems of slow learning convergence, easy to fall into local minima and uncertain network structure of the traditional BP algorithm, and further optimize the weights and thresholds, resulting in optimal configuration of network parameters, minimal prediction error of the test set and higher prediction accuracy. The GA-PB neural network refers to a BP neural network in which the initial weights and thresholds from the input layer to the implicit layer and from the implicit layer to the output layer are encoded and concatenated into a body. To be specific, the individuals are screened by the selection, crossover, and change operations in the genetic algorithm, and the good individuals are screened out and the poorly adapted individuals are eliminated in accordance with the fitness values of the fitness function. Furthermore, the screened good individuals are fed into the BP neural network as the initial weights and thresholds of the BP algorithm [41,45], and the learning process is expressed as follows:
Using the GA algorithm to filter the connection right parameters $W_{ij}$ for initializing i and j and the threshold parameters $\theta_{j}$ for node j;Read in the pre-processed training samples $\left\{X_{PL}\right\}$ and $\left\{Y_{PK}\right\}$;Taking $I_{pj}$ as the output of the node i and the input of the node j, the outputs of the nodes at the respective level are expressed as follows:(9)$$O_{pj}=f\sum_{i}(w_{ij}I_{pj}-\theta_{j})$$The nodal error signals $\delta_{pk}$ and $O_{pi}$ for the output and hidden layers are expressed as follows:(10)$$\left\{ \begin{array}{l} \delta_{pk}=O_{pk}(y_{pk}-O_{pk})(1-O_{pk})\\ O_{pi}=O_{pi}(1-O_{pi})\sum_{i}\theta_{pi}w_{ij} \end{array}\right.$$Back-propagation, correction of weights and calculation of residuals.
(11)$$\left\{ \begin{array}{l} W_{ij}(t+1)=\alpha \delta_{pi}O_{pi}+W_{ij}(t)\\ E_{p}=\sum_{p}\sum_{k}\frac{{\left(O_{pk}-Y_{pk}\right)}^{2}}{2} \end{array}\right.$$


Based on the GA-BP neural network algorithm, the factors for glacier morphology change, slope, slope aspect, elevation, glacier albedo, surface organic carbon density, and rainfall, which have been analyzed by grey correlation [46], were taken as the input layers in this study. Moreover, the number of implied layers was determined using the empirical equation [41], and the glacier cumulative morphological variables were taken as the output layers to predict the glacier morphology of the Meili Mountain (Figure 8).

## 4. Results

### 4.1. Glacier Deformation Information Acquisition and Spatial and Temporal Separation Analyses of Change Characteristics

Glacier deformation has been divided into negative and positive deformation. To be specific, negative deformation includes glacier ablation, downslope movement, and thickness reduction, while positive deformation includes glacier accumulation, upslope movement, as well as thickness increase [7]. Sentinel-1 data from January 2020 to December 2021 in the study area were adopted to obtain the line-of-sight directional deformation (Figure 9a,b), azimuthal deformation (Figure 9c,d), time series deformation rate (Figure 9e), cumulative deformation (Figure 9f), 3D deformation field (Figure 9g–i), and longitudinal and cross-sectional deformation characteristics (Figure 10 and Figure 11) of the Meridian Mountain glacier. The results indicated that the spatial variability of glacier deformation in the Meili Mountain was evident, with a north–south distribution along the Meili Snow Mountain range, with glacier ablation concentrated in the southern and northern glacier boundaries and extensive accumulation in the central part of the glacier, i.e., at higher elevations. In the study period, the maximum deformation of the Meili Mountain glacier reached 135.19 mm in the line-of-sight direction, −48.76–131.33 mm in the azimuthal direction, and −212.16 mm in the cumulative time series. Moreover, the average annual deformation rate ranged from −94.62 to 75.96 mm/year. The maximum deformation in the vertical, north–south, and east–west directions after 3D decomposition reached −125.63 mm, respectively. The AA’ deformation characteristics of the longitudinal section along the north–south trend of the mountain range indicated that the northern and southern edges of the Meili Mountain glacier primarily showed a melting trend, whereas the central area primarily showed glacier accumulation. The longitudinal section was divided into the northern ablation zone, the central accumulation zone, and the southern ablation zone based on the deformation characteristics of the longitudinal section to comprehensively analyze the deformation trend of the Meridian Mountain glacier (Figure 10). The deformation characteristics of the central accumulation zone was correlated with the altitude to a certain extent, and the accumulation maximum was located at the middle of the CC’ profile. The deformation characteristics of the southern melting zone reached the maximum in the middle of the zone and tended to increase along both sides of the profile boundary to the middle of the profile. The deformation characteristics of the southern melting zone reached their maximum in the middle of the region and tended to increase along both sides of the profile boundary to the middle of the profile.

The major reason for the inconsistency in the deformation trends of the north–south deformation components is that the InSAR deformation observations are extremely insensitive to the deformation in the north–south direction due to the limitations of the existing SAR satellite’s polar orbit flight and side-view imaging mode [34]. Several eponymous deformation points were randomly selected for regression analysis of the northern melt zone, the central accumulation zone, and the southern melt zone using different methods of cross-validation with the same orbital dataset (Figure 12) to further quantify the accuracy of the results. The cross-validated R2 of the D-InSAR and SBAS-InSAR homonymous points were 0.87, 0.83, and 0.76, respectively, indicating that the glacier deformation built based on the D-InSAR and SBAS-InSAR techniques has a good correlation on the time scale, thus confirming the accuracy and validity of the InSAR deformation results in this study.

### 4.2. Selection of Glacier Deformation Response Characteristics Factors and Analysis of Factor Drivers

Relevant research has suggested [5,7,8,15,17] that the main factors for the climate and environment of mountain glaciers on the Qinghai-Tibet Plateau ice sheet and thus glacier morphology [47,48] include altitude, slope, slope orientation, rainfall, as well as glacier albedo. In addition, changes in surface organic carbon density due to human and natural activities can cause regional climatic and environmental changes [49], thus affecting glacier morphology. The change in mountain glacier morphology is an extremely complex process, and it is affected by spatial and temporal differentiation characteristics. Moreover, the melting and accumulation of mountain glaciers are highly regional and seasonal. Grey correlation analysis was conducted to the northern melting zone, the central accumulation zone, and the southern melting zone to correlate the six factors with the morphological change values, so as to gain insights into the contribution of the above factors to the melting and accumulation of mountain glaciers. The results indicated that the grey correlation values of the factors in the northern melting zone included glacier albedo (0.973) > slope (0.938) > surface organic carbon density (0.936) > elevation (0.935) > aspect (0.900) > rainfall (0.755). The grey correlation values of the factors in the central accumulation zone included glacier albedo (0.991) > surface organic carbon density (0.988) > slope (0.979) > elevation (0.970) > aspect (0.952) > rainfall (0.844). In addition, the grey correlation values of the factors in the southern melting zone were glacial albedo (0.980) > surface organic carbon density (0.963) > slope (0.954) > aspect (0.947) > elevation (0.946) > rainfall (0.811), as listed in Table 2.

As depicted in the above table, there was a high correlation between elevation, slope, slope orientation, glacier albedo, surface organic carbon density, and rainfall with the morphological change values of the mountain glaciers in the Meili Snow Mountain, thus suggesting that the above factors significantly affect morphological changes (e.g., melting and accumulation of mountain glaciers) and are highly correlated with the morphological changes in the mountain glaciers in the Meili Snow Mountain. Furthermore, there were significant differences in the driving forces of the morphological changes in the mountain glaciers by different factors.

### 4.3. GA-BP Model Construction and Glacier Deformation Prediction

Based on the GA-BP neural network algorithm, the melting region was selected as the study area for glacier morphological change prediction in this study. Factor analysis was conducted on elevation, slope, aspect, surface organic carbon density, glacier albedo and rainfall to gain insights into whether the glacier morphology change factors selected in Section 4.2 are suitable as factors for surface sedimentation neural network prediction. The results indicated that the optimal test statistic was achieved when a combination of six factors was selected. The GA-BP neural network model was built by integrating the six selected factors with the cumulative shape variables of the melting zone, as well as by setting the number of nodes in the input layer to 6, the number of nodes in the hidden layer to 9, and the number of nodes in the input layer to 1.756. The points selected by the GA algorithm were employed as the training samples, and 56 of them were predicted. The network functions and training parameters of the respective layer of the GA-BP model built are listed in Table 3.

A standard BP neural network model was selected for comparison experiments with the prediction accuracy of the GA-BP neural network model in glacier surface deformation monitoring to verify the validity and feasibility of the model used in this study, and to explore the effect of the GA algorithm on the performance of the BP neural network. In the above experiments, the network functions, training parameters, and deformation datasets of the respective layer were controlled, and 700 randomly selected deformation data from the melting area were employed as the training samples. The remaining 56 deformation points were predicted, and the prediction results are presented in Figure 13. As depicted in the figure, the standard BP neural network, the GA-BP neural network model, and the InSAR monitoring results were generally consistent, and the predicted and actual curves were approximately the same despite the existence of some outlier anomalies. Neural network and GA-BP neural network algorithm were 0.67 and 0.86, respectively. Moreover, the two network models showed their convergence with the optimal number of iterations of 105 and 30, respectively (Figure 13a,b), with the average relative errors of 1.12% and 1.17%, and the root mean square errors of 9.58 and 10.38 mm, respectively, as listed in Table 4.

## 5. Discussion

### 5.1. Characterization of the Spatial Distribution of Glacial Deformation in the Meili Snow Mountain

To gain insights into the difference in deformation characteristics between the ablation and accumulation areas, the cumulative deformation of the Meili Mountain glaciers was divided (Figure 14), with the maximum deformation of −81.36 mm in the northern melting area, 161.67 mm in the central accumulation area, as well as −212.63 mm in the southern melting area. The reason for this is that mountain glaciers extend along gravity and the glacier terminus is relatively low in elevation, such that the glacier terminus exhibited strong melting characteristics, decreasing in thickness and moving downwards. In addition, the larger area in the central part of the glacier showed accumulation, while the smaller glaciers in the north and south showed glacier ablation, consistent with previous studies which show that small glaciers are more responsive to climate change [7]. Zonal statistics also suggest that mountain glacier deformation evolves along the mountain range in a similar pattern to rock glaciers [50]. To be specific, the glacier deformation rates were the slowest in the northwest and the fastest in the southeast of the Meili Snow Mountain range.

### 5.2. Analysis of the Temporal Evolution of Glacier Deformation in the Meili Snow Mountain

Mountain glacier deformation is characterized by complex causes and diverse types. since glacier deformation can be based on the assumption that it comprises seasonal non-linear deformation and long-term linear subsidence trends [15], this study selected characteristic points for time series analysis of the accumulation zone and melting zone and fitted them in accordance with the principle of least squares (Figure 15a,b). The results indicated that the accumulation and melting zones show different deformation trends due to differences in spatial distribution. Due to the change in elevation gradient, the deformation in the central part of the accumulation zone is more intense compared with the marginal area. The difference in deformation characteristics between the northern and southern melting areas is relatively small, but the deformation quantity in the northern melting area is larger than that in the southern melting area, which is most likely related to the topography of the study area. The northern section of the Meili Mountain glacier is wide and thick, with less dramatic topographic relief and incomplete glacier development, while the southern section of the glacier is narrow and has dramatic topographic relief, forming a relatively mature alpine and glacial landscape. Moreover, Figure 15b also illustrates that the difference of mountain glacier deformation between wet and dry seasons was significant, and rainfall in the wet season had a reciprocal effect on the accumulation area and the melting area, thus promoting glacier uplift in the accumulation area while accelerating glacier ablation in the melting area. The linear trend of glacier melting in the north and south regions was nearly −116 mm/a, similar to the results of Wenjie Du et al. [15], whereas there were differences in quantitative values, which may arise from the different penetration abilities of L-band and C-band SAR data on glaciers and granular snow. In order to further analyze the influence of rainfall changes in wet and dry seasons on glacier deformation time series, a superposition analysis was conducted between rainfall data in the study period and deformation time series curves of characteristic points (Figure 15b). It can be seen that there is a significant correlation between glacier deformation and rainfall. Two large precipitation events between April and May 2020, and between July and August 2021 both increased the glacier deformation in the accumulation and melting areas, while the glacier deformation was relatively gentle in the dry season with less rainfall. The deformation time series of characteristic points and rainfall time series do not show a synchronous change relationship, which indicates that the morphological change of Meili Mountain glacier has a certain hysteretic response to rainfall, and indicates that the deformation of mountain glacier is affected by many aspects, and the deformation process is relatively complex [7]. It is proved that the prediction model is scientific and reasonable.

### 5.3. Analysis of the Drivers of Glacier Deformation Factors in the Meili Mountain

The results of the grey correlation values in the previous Section 3.2 indicated that glacier albedo contributed the most to the driving force of systematic morphological changes in glacier melting and accumulation zones, thus suggesting that albedo changes are the main factor driving mountain glacier morphology, consistent with the findings of Zhang Y et al. [13]. Glacier albedo is capable of characterizing the ratio of reflected solar radiation to incident solar radiation at the glacier surface, and decreasing and increasing glacier albedo serves as a direct factor for glacier melting and accumulation. In addition, the results of the grey correlation value ranking (Table 2) suggest that surface organic carbon density significantly drives glacier morphological changes in the Meridian Mountains, which is an essential complement to the analysis of glacier deformation. In particular, black carbon (BC) in snow and ice exhibits a strong light absorption capacity [51], and changes in carbon content directly contributes to changes in temperature, thus affecting the morphological changes in the Meridian Mountains glaciers. Grey correlation values indicated that elevation factors were second only to glacier albedo, surface organic carbon density and slope as drivers of glacier morphology in the Meili Mountain. To investigate whether mountain glaciers meet the above rule, the statistical analysis of the shape variables of Meili Mountain glaciers according to elevation (Figure 16) indicated that the elevation value at 4500 m was the characteristic point of divergence for the change in shape gradient of Meili Mountain glaciers; the absolute value of shape variables was larger below 4500 m, and the absolute value was smaller above 4500 m.

### 5.4. Evaluation of the Prediction Performance of the GA-BP Neural Network Algorithm

In this study, a hybrid prediction model is developed from a multi-factor perspective using a feed-forward neural network trained by a combination of genetic algorithm and back-propagation algorithm. Unlike previous prediction models using mainly historical deformation data, this method has better prediction results and lower prediction errors. The decision coefficients, convergence optimal number of iterations, MRSE, and mean relative error (Table 4) verify that the GA-BP neural network used in this study can effectively overcome the problems of the standard BP algorithm relying on initial weights and thresholds and lower accuracy, and effectively improve the network prediction performance and accuracy, indicating that the GA-BP neural network algorithm has better reliability in predicting mountain glacier deformation trends.

## 6. Conclusions

Mountain glaciers are effective in indicating the regional-scale hydrological cycle and climate change. To solve the limitations in the monitoring and prediction of mountain glacier deformation, this study proposed a method to monitor and predict glacier deformation from a multi-factor perspective in combination with InSAR technology and GA-BP neural network. First, the time series of mountain glacier deformation and 3D deformation field were built by combining SBAS-InSAR, D-InSAR, and MAI techniques, and the spatial distribution characteristics and temporal evolution pattern of the glacier deformation in the Meili Mountain were analyzed. Subsequently, the relationship between mountain glacier deformation variables and climate and environmental factors was developed and employed as a training sample to optimize the BP neural network using genetic algorithm (GA). The following conclusions were obtained by predicting mountain glacier deformation:

(1) The temporal evolution of glacier deformation in the Meili Mountain is more significant, with the maximum deformation of 135.19 mm in the differential interference line-of-sight direction, −48.76~131.33 mm in the azimuthal direction, the cumulative maximum deformation of −212.16 mm in the time series, and an average annual deformation rate from −94.62 to 75.96 mm/year, with the maximum deformation of −125.63 mm, −77.03 mm and 107.98 mm in the vertical, north–south, and east–west directions, respectively after 3D decomposition. The maximum deformation change in the vertical, north–south, and east–west directions reaches −125.63 mm, −77.03 mm, and 107.98 mm, respectively;

(2) The spatial distribution of glacier deformation in the Meili Mountain exhibits strong divergence, with mountain glacier deformation being the slowest in the northwest and the fastest in the southeast along the mountain range, and with smaller mountain glaciers showing a stronger response to climate change;

(3) Surface organic carbon density is a vital driver of morphological change in the Meridian Mountains, and it serves as an essential complement to the analysis of glacier morphology. The strong light absorption capacity of black carbon (BC) at the ice sheet surface directly affects ice sheet temperature, thus having an effect on the morphological change in the Meridian Mountains glaciers;

(4) The 4500 m elevation value is the point of divergence characteristic of the change in the gradient of deformation of the Meridian glacier. The absolute value of the deformation variable is larger for elevations below 4500 m, and that is smaller above 4500 m;

(5) The coefficient of determination, average relative error, and root mean square error of the optimized GA-BP neural network algorithm based on the genetic algorithm for predicting the glacier deformation in the Meili Mountain are 0.86 mm, 1.12%, and 10.38 mm respectively, thus suggesting that the GA-BP neural network used in this study can effectively overcome the problems of the standard BP algorithm relying on initial weights and thresholds and low accuracy, and effectively improve the network prediction. This study reveals that the GA-BP neural network is capable of effectively overcoming the problems of the standard BP algorithm (e.g., reliance on initial weights and thresholds and low accuracy) and effectively enhancing the network prediction performance and accuracy, thus providing an effective and reliable prediction of glacier deformation.

## Figures and Tables

**Figure 1 sensors-22-08350-f001:**
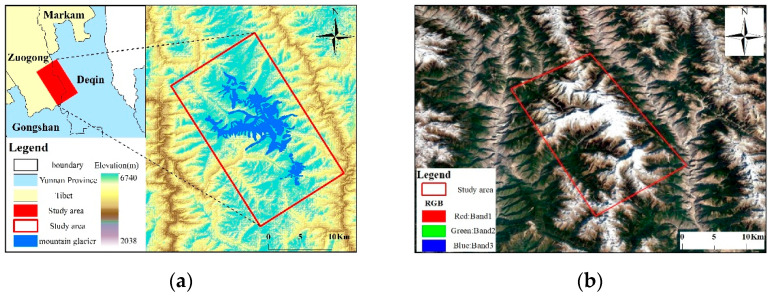
Location study area location. (**a**) Location and topography of the study area. (**b**) High resolution remote sensing image map of the study area, true color composite from R, G, and B bands.

**Figure 2 sensors-22-08350-f002:**
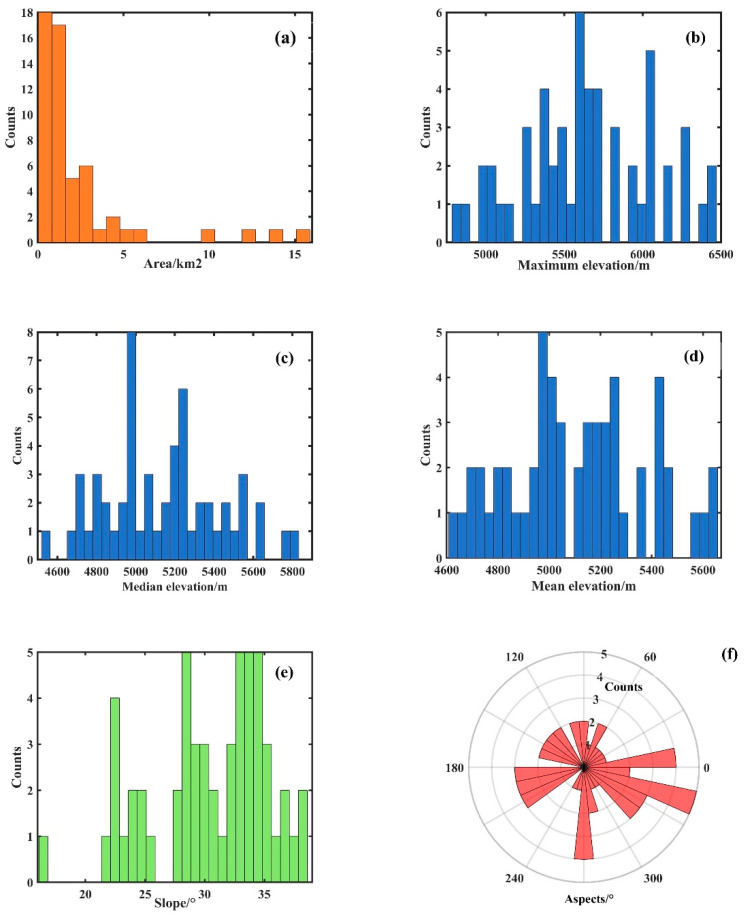
(**a**) Statistical map of mountain glacier area, (**b**–**d**) statistics of maximum, median and mean elevation of mountain glaciers, (**e**) statistical distribution of mean slope of mountain glaciers, (**f**) statistical distribution of mean aspects of mountain glaciers.

**Figure 3 sensors-22-08350-f003:**
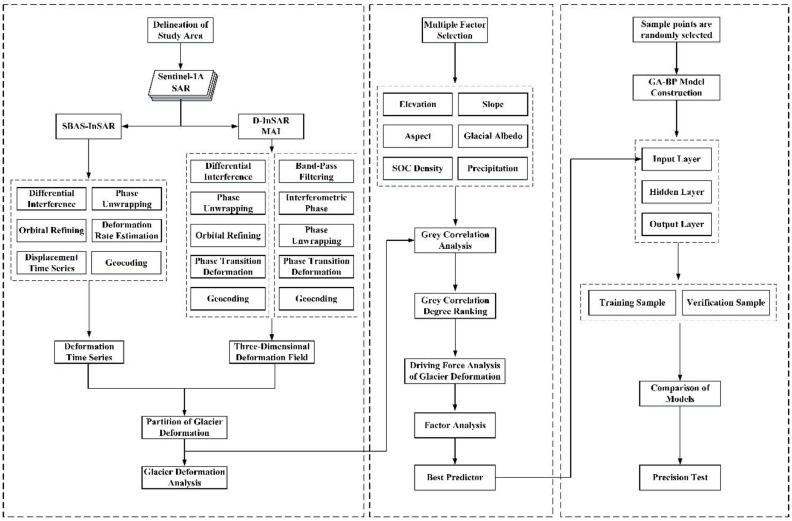
Technical flow chart.

**Figure 4 sensors-22-08350-f004:**
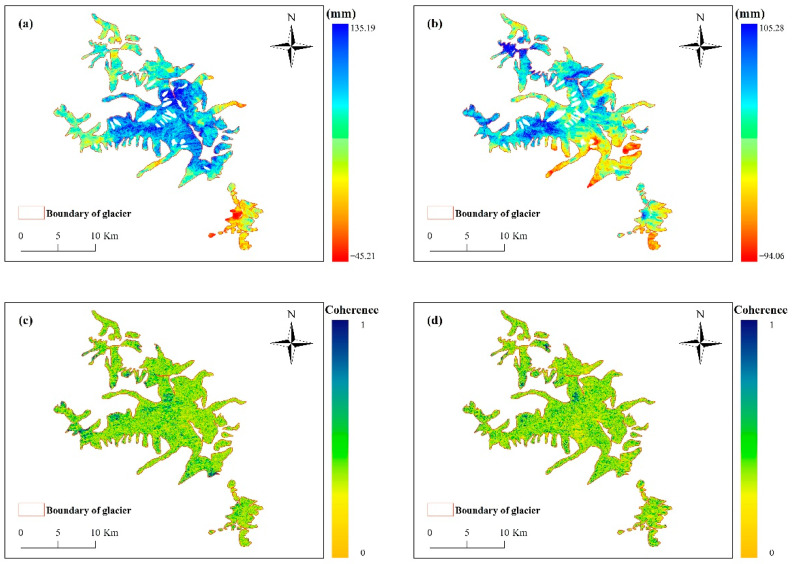
Deformation results in the direction of the DInSAR line-of-sight, (**a**) is the result of ascending orbit, (**b**) is the result of descending orbit, (**c**) is the coherence of ascending orbit, (**d**) is the coherence of descending orbit.

**Figure 5 sensors-22-08350-f005:**
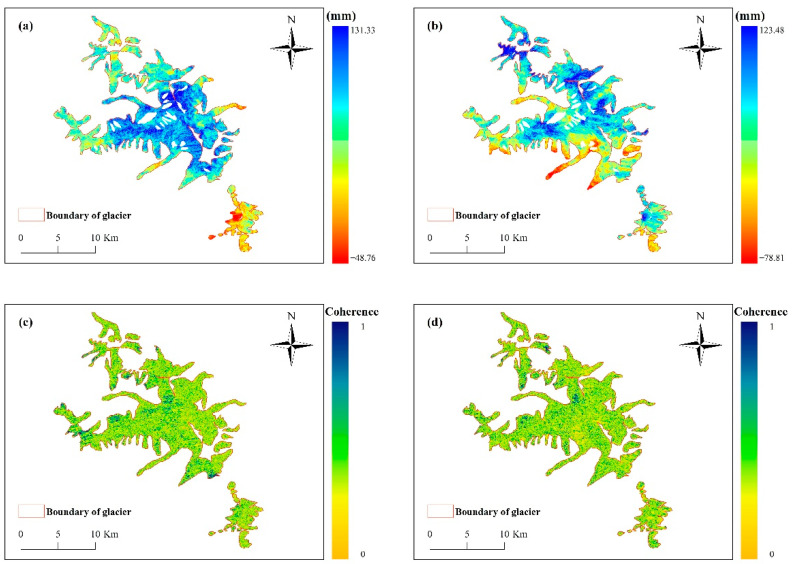
Azimuthal deformation results of the MAI technique, (**a**) is the result of ascending orbit, (**b**) is the result of descending orbit, (**c**) is the coherence of ascending orbit, (**d**) is the coherence of descending orbit.

**Figure 6 sensors-22-08350-f006:**
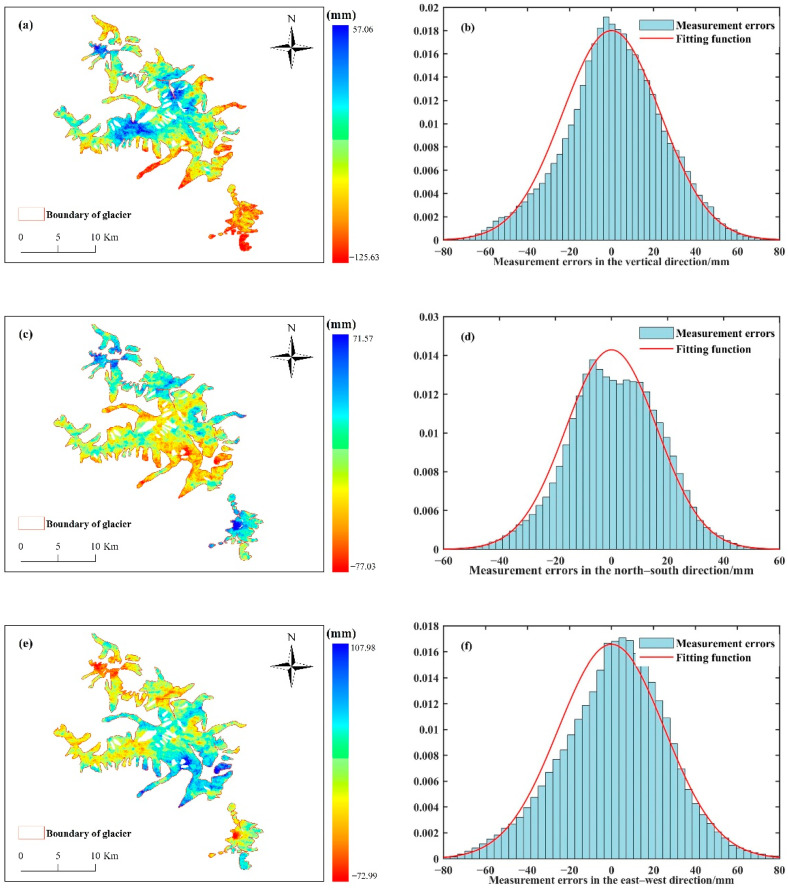
Three-dimensional deformation field of the Meili Mountain glacier, (**a**) vertical component, (**b**) measurement errors in the vertical direction, (**c**) north–south component, (**d**) measurement errors in the north–south direction, (**e**) east–west component, (**f**) measurement errors in the north–south direction.

**Figure 7 sensors-22-08350-f007:**
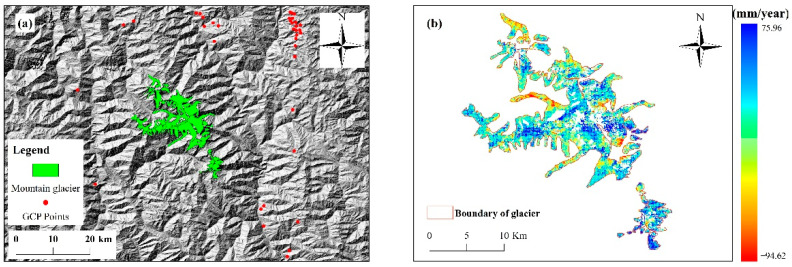
Time series deformation results for the Meili Mountain glacier, (**a**) GCP point, (**b**) deformation rates, (**c**) deformation variables, (**d**) average coherence of SAR images.

**Figure 8 sensors-22-08350-f008:**
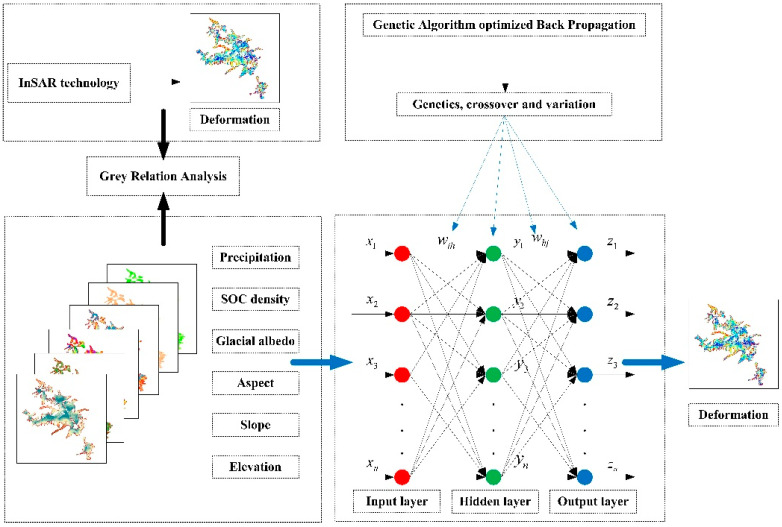
Flow chart of GA-BP neural network.

**Figure 9 sensors-22-08350-f009:**
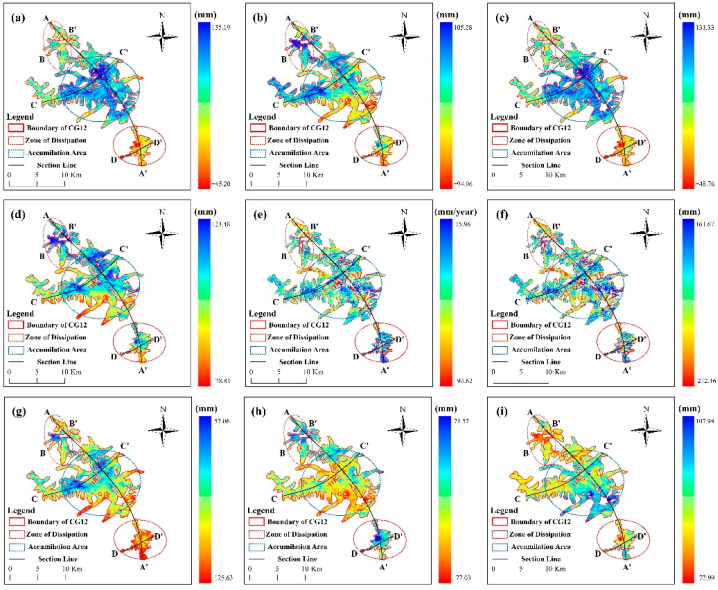
Meili Mountain glacier accumulation and ablation zone deformation variables (“AA’” is the vertical section of the mountain glacier, ”BB’”, “CC’” and “DD’” are transects of the glacier accumulation and ablation zone, respectively.), (**a**) ascending orbit DInSAR results, (**b**) descending orbit DInSAR deformation results, (**c**) ascending orbit MAI results, (**d**) descending orbit MAI results, (**e**) SBAS-InSAR deformation rates, (**f**) SBAS-InSAR deformation variables, (**g**) 3D deformation vertical results, (**h**) 3D deformation north–south results, (**i**) 3D deformation east–west results.

**Figure 10 sensors-22-08350-f010:**
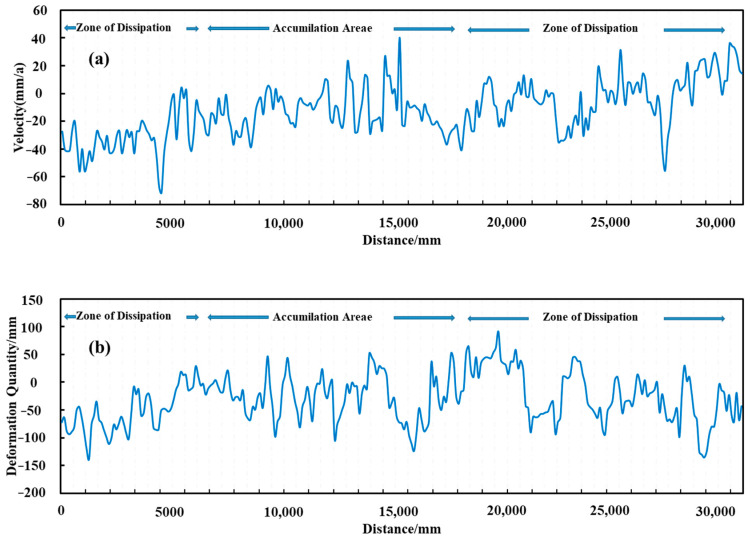
Longitudinal profile deformation characteristics; (**a**) is the longitudinal profile deformation rate, (**b**) is the cumulative deformation variable.

**Figure 11 sensors-22-08350-f011:**
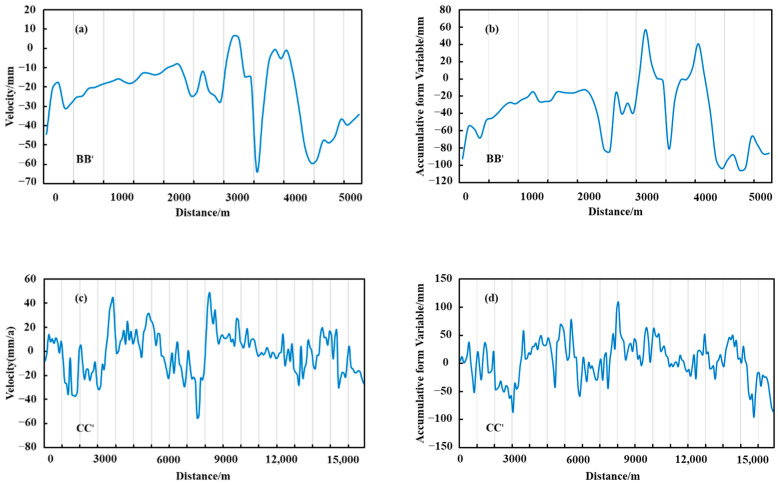
Cross-sectional deformation features. (**a**,**c**,**e**) the Cross-sectional deformation rate, (**b**,**d**,**f**) the cumulative deformation variable.

**Figure 12 sensors-22-08350-f012:**
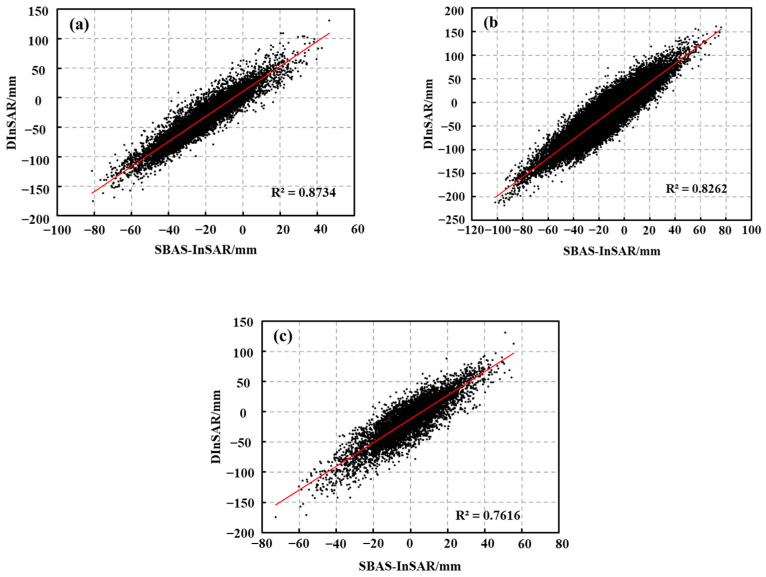
Results of cross-validation of glacier deformation features in the Meili Mountain, (**a**) for the northern melt zone, (**b**) for the central accumulation zone, and (**c**) for the southern melt zone.

**Figure 13 sensors-22-08350-f013:**
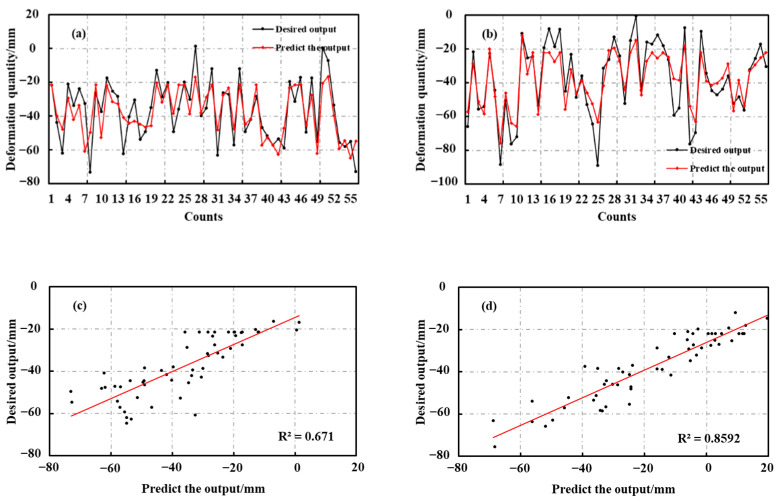
Comparison of the prediction results of BP and GA-BP neural network algorithms. (**a**) prediction results of BP neural network, (**b**) prediction results of GA-BP neural network, (**c**) R^2^ of BP neural network, (**d**) R^2^ of GA-BP neural network, (**e**) number of iterations of BP neural network, (**f**) number of iterations of GA-BP neural network.

**Figure 14 sensors-22-08350-f014:**
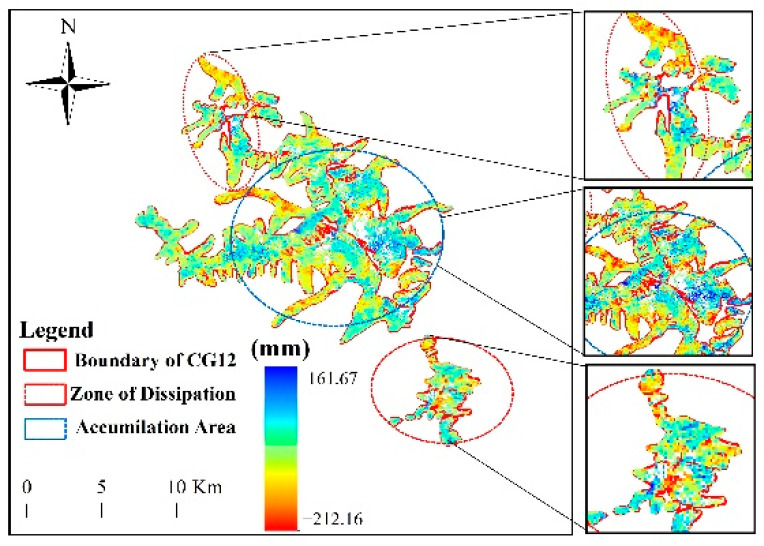
Statistics of glacier deformation zoning in the Meili Snow Mountain.

**Figure 15 sensors-22-08350-f015:**
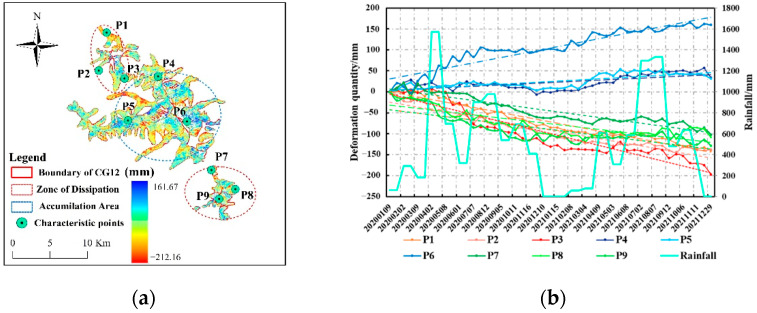
(**a**) Distribution of deformation characteristic points, (**b**) time series curves of characteristic points of glacier deformation in the Meili Snow Mountain.

**Figure 16 sensors-22-08350-f016:**
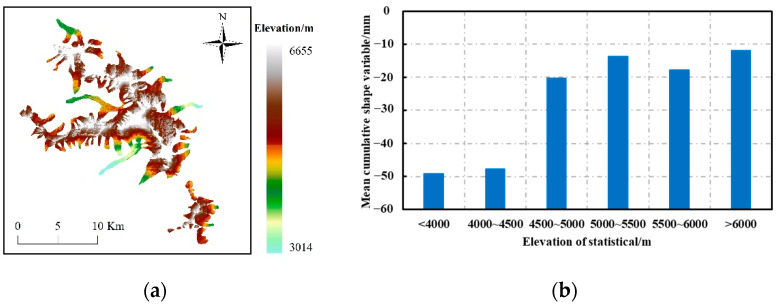
(**a**) Glacier elevation (m) in the study area. (**b**) Statistics of measured glacier deformation at different elevations.

**Table 1 sensors-22-08350-t001:** Data sources and specifications.

Data Name	Data Phase	Data Type	Data Scale	Data Source
Sentinel-1 radar image	2020-01–2021-12	Raster	5 m × 20 m	ESA
ALOS DEM	-	Raster	30 m	JAXA
Elevation	-	Raster	30 m	JAXA
Slope	-	Raster	30 m	JAXA
Aspect	-	Raster	30 m	JAXA
Glacial albedo	2020-01–2021-12	Raster	30 m	Landsat 8
SOC density	2018	Raster	250 m	FAO
Precipitation	2020-01–2021-12	-	-	PPS
Google Maps imagery	2021	Raster	0.2 m	Google Earth

**Table 2 sensors-22-08350-t002:** Grey correlation factors for melting and accumulation zones.

	Northern Melting Zone	Middle Accumulation Region	Southern Melting Zone
Elevation	0.935	0.970	0.946
Slope	0.938	0.979	0.954
Aspect	0.900	0.952	0.947
Glacial albedo	0.973	0.991	0.980
SOC density	0.936	0.988	0.963
Precipitation	0.755	0.844	0.811

**Table 3 sensors-22-08350-t003:** Network functions and training parameters for the respective layer of the GA-BP model.

Network Functions for Each Layer	Setting Functions	Name of the Parameter	Setting Values
Activation function for the hidden layer	tandig	Training times	10,000
		Learning rate	0.01
Activation function of the input layer	purelin	Evolutionary algebra	40
		Population size	20
Training function	trainlm	Crossover probabilities	0.2
		Probability of variation	0.1

**Table 4 sensors-22-08350-t004:** Comparison of the accuracy of BP and GA-BP results.

	BP	GA-BP
R^2^	0.67	0.86
Number of iterations	105	30
RMSE	10.38	9.58
MAPE	1.17%	1.12%

## Data Availability

Not applicable.

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
