# Peer review of "Monitoring and Prediction of Glacier Deformation in the Meili Snow Mountain Based on InSAR Technology and GA-BP Neural Network Algorithm"

_sensors, 2022, doi:10.3390/s22218350_

Round 1
Reviewer 1 Report
Overview
The authors utilized InSAR and Multiple Aperture Interferometry (MAI) processing of both ascending and descending Sentinel-1 data spanning from January 2020 to December 2021 to obtain three-dimensional deformation of the glaciers in Meili mountain, southwestern China. Taking those Sentiel-1 observations as input, the authors used a model based on back propagation neural network (GA-BP) to predict future movement of the glaciers in the same region.
This manuscript is overall very interesting in terms of the complex methods utilized for obtaining SAR measurements and for projecting future glacier motions. The high mountain areas are often challenging for retrieving continuous high-accuracy SAR measurements owing to the steep hillslopes and seasonal snowfall. Considering these challenges, this manuscript presented quite interesting results and could potentially make great contributions to this field. More detailed information of multiple parts, especially the data processing methodology and the quality of the SAR observations, is required to better support the results and conclusions in this manuscript.
I therefore recommend “major revision” of this manuscript before reconsideration for publication.
Major comments
The methodology: This manuscript well described the theoretical framework of the used methods, such as MAI and GA-BP. How were these methods implemented in this study? Did the authors write their own codes? If a specific package was used instead, please provide the related information as a reference.
The InSAR and MAI results: In Figure 4 and Figure 5, it is very difficult to tell whether those SAR measurements are showing glacier movement or just noise. Most areas seemed to show “deformation signals”. Does that mean this region is dominated by actively moving glaciers? Are there ground-truth data to help verify those SAR measurements? What did InSAR capture over the mountains outside the target glacier area? That might help explain the accuracy of the SAR measurements, because the theoretical deformation of those non-glacier mountains is zero. A coherence map can also help show the potential quality of the SAR observations.
Figure 16a indicates that this area contains many steep slopes. How did those steep slopes affect the side-looking SAR’s observation capability? It seems in Figures 4,5 that this area was not impacted by the layover and shadowing effects, which is curious.
If assuming the glacier movement directions follow surface topography, both InSAR and MAI have highly variable sensitivity to their motions and thus affecting the ability to invert for the 3D movement vectors. For example, the MAI can only measure deformation along the flight direction. A measurement uncertainty map is necessary for the 3D measurements.
The time-series measurements: Where are the locations of those “characteristic points” shown in Figure 15? It would be helpful to show the motion time series of multiple glaciers in the study area and analyze how much their temporal patterns differed and why.
The GA-BP analysis: The evaluation of potential environmental drivers primarily relied on the spatial variability of multiple glaciers in the study area, which makes sense for factors such as elevation, slope, and aspect. However, as this study also obtained deformation time series of those glaciers, it would be valuable to use the time-series data to evaluate the impacts of factors (e.g., rainfall) that might vary significantly in time.
Minor Comments
Line 19: what does “relatively single” mean?
Lines 18 – 22: This sentence is very long and its meaning is unclear. Please rephrase.
Line 21: add the full name of GA-BP at its first appearance in the manuscript.
Line 47: “China’s glaciers” --> The glaciers in China.
Line 141: “combining” --> that combines
Line 155: the total area is only “1000 m2”?
Figure 2: These figures are a bit confusing. It might be better to show the probability distributions (e.g., the “area” as x-axis and the “counts” as y-axis).
Equation (1) and (6): Why the atmospheric contributions were not considered in Equation (1) and orbit errors were not considered in Equation (6)?
Line 313: How was the InSAR coherence of two SAR images spanning 60 days in the snow season?
Reviewer 2 Report
The authors propose a way to monitor and predict the glacier deformation in Meili Snow Mountain by applying InSAR technologies and GA-BP neural networks. Several comments are given as follows:
1. The authors used different InSAR techniques (D-InSAR, MAT-InSAR, SBAS-InSAR) to have the deformation information of the study areas. The deformation results by applying other InSAR techniques are illustrated in Figure 8. From the results, it is difficult to find the differences. Can authors offer a better way to describe the differences?
2. GA-BP is one of the artificial neural networks. Why are the authors interested in applying the GA-BP network to predict the deformation information? The authors are supposed to provide more descriptions.
3. The authors apply 700 randomly selected deformation data as the training data set and 56 data points to evaluate the predictions. The study areas contain 55 glaciers and 128.54 km2. Those selected data points are few with comparing the study areas. Those selected points are supposed to lie over the 55 glaciers to reflect the true deformation information. Randomly selected points may not fit the problem.
4. The authors did a lot of work, and a better presentation is needed.
Reviewer 3 Report
1) I think the abstract need to rewrite. For example, the main statement and rationale of the manuscript from lines 18-22 are hard to follow (passive voice). And authors also put too much information in the results part. The abstract therefore needs to rewrite and should be balanced for all parts (e.g., rationale, materials, methods, results, conclusion, discussion/future work).
2) The first paragraph raises the rationale for mountain glacier melting. But why only mention China’s glaciers? How about the information on different mountain glaciers around the world?
3) Please provide the references for the statement of "mathematical and statistical models" lines 125-127
4) What does GA-BP stand for? Genetic Algorithm optimized Back Propagation (GA-BP) neural network?
5) figure 3 quality is bad; it is hard to read the flow chart. Can authors also provide a flowchart for the integration of InSAR and GA-BP neural network? What are the input and output vectors of the neural network?
6) Figure 12 actually shows the model as well as input-output vectors. Why place this section in the result part? It is the main integration of InSAR and GA-BP.
7) Have the authors compared the proposed technique with different techniques such as [1,2]?
8) Please perform the spelling and grammar check.
[1] Du, W.; Ji, W.; Xu, L.; Wang, S. Deformation Time Series and Driving-Force Analysis of Glaciers in the Eastern Tienshan Mountains Using the SBAS InSAR Method. Int. J. Environ. Res. Public Health 2020, 17, 2836.
[2] Zhang, T.; Zhang, W.; Cao, D.; Yi, Y.; Wu, X. A New Deep Learning Neural Network Model for the Identification of InSAR Anomalous Deformation Areas. Remote Sens. 2022, 14, 2690. https://doi.org/10.3390/rs14112690
Round 2
Reviewer 1 Report
Overview
The authors have addressed most of the questions from the first round of the review. There are a few minor points that can help improve the clarity of this manuscript.
Minor Comments
Line 201: Please elaborate on what “48-view” means.
Line 267: Please specify how the coherence was obtained. Are those the averaged InSAR coherences spanning the observation period?
Line 287: same as above.
Figure 15: The glacier movement might correlate with seasonal temperature variations. The deformation time series in Figure 15b seem to be very noisy and barely show any seasonal signals. Where is the location of the reference area for the InSAR measurements?
Reviewer 3 Report
The authors have addressed my comments.
